# How Does Time Use Differ between Individuals Who Do More versus Less Foodwork? A Compositional Data Analysis of Time Use in the United Kingdom Time Use Survey 2014–2015

**DOI:** 10.3390/nu12082280

**Published:** 2020-07-30

**Authors:** Chloe Clifford Astbury, Louise Foley, Tarra L. Penney, Jean Adams

**Affiliations:** 1MRC Epidemiology Unit, Centre for Diet and Activity Research (CEDAR), Institute of Metabolic Science, Cambridge Biomedical Campus, University of Cambridge, Cambridge CB2 0QQ, UK; tpenney@yorku.ca (T.L.P.); jma79@medschl.cam.ac.uk (J.A.); 2Global Health Program, Faculty of Health, York University, Toronto, ON M3J 1P3, Canada; 3Global Diet and Activity Research Group and Network (GDAR), MRC Epidemiology Unit, Institute of Metabolic Science, Cambridge Biomedical Campus, University of Cambridge, Cambridge CB2 0QQ, UK; lf354@medschl.cam.ac.uk

**Keywords:** foodwork, home food preparation, compositional data analysis, time-use data

## Abstract

Background: Increased time spent on home food preparation is associated with higher diet quality, but a lack of time is often reported as a barrier to this practice. We compared time use in individuals who do more versus less foodwork (tasks required to feed ourselves and our households, including home food preparation). Methods: Cross-sectional analysis of the UK Time Use Survey 2014–15, participants aged 16+ (*N* = 6143). Time use over 24 h was attributed to seven compositional parts: personal care; sleep; eating; physical activity; leisure screen time; work (paid and unpaid); and socialising and hobbies. Participants were categorised as doing no, ‘some’ (<70 min), or ‘more’ foodwork (≥70 min). We used compositional data analysis to test whether time-use composition varied between these participant groups, determine which of the parts varied between groups, and test for differences across population subgroups. Results: Participants who spent more time on foodwork spent less time on sleep, eating, and personal care and more time on work. Women who did more foodwork spent less time on personal care, socialising, and hobbies, which was not the case for men. Conclusion: Those who seek to encourage home food preparation should be aware of the associations between foodwork and other activities and design their interventions to guard against unintended consequences.

## 1. Introduction

Observational evidence suggests that higher frequency of making [1,2,3,4,5] and eating [4,6,7] home-prepared meals, while not a prerequisite for a healthy diet [8], is associated with improved dietary intake and health outcomes. Substantial energy has been devoted to understanding the modifiable determinants of home food preparation and increasing food preparation in households [9,10,11,12,13].

Existing research has explored how time spent on ‘foodwork’, the tasks required to feed ourselves and our households, including preparation, shopping, cleaning up, and washing dishes [14], has evolved in different countries [15,16,17]. A study of time use in the UK found that time spent on food preparation decreased by 16 min between 1975 and 2000, while participation in home food preparation had increased (from 75% to 83% of the sample), a change driven principally by the increasing participation of men in this task [18].

Increased time spent on foodwork has been shown to positively impact diet [19]. More time spent on foodwork may represent a higher frequency of preparing meals at home or a particular kind of home food preparation, preparing food ‘from scratch’ [20] or from unprocessed or minimally processed ingredients, which has been posited to be particularly important to achieving high diet quality [21].

Beyond its potential association with diet quality, time is an important dimension in understanding home food preparation due to the frequency with which a lack of time is cited as a barrier and the importance of time and convenience in structuring food practices and attitudes towards them [22]. In numerous studies, participants report preparing food at home less often than they would like because they feel they lack the time [20,23,24,25,26,27,28].

As interventions designed to increase the frequency of home food preparation and increase cooking ‘from scratch’ are being implemented [10,11,13], this key barrier is worthy of further exploration. While income-related barriers to healthy eating have been explored and, to some extent, integrated into theory and intervention design, evidence assessing associations between time scarcity and healthy eating behaviours is more limited, and few interventions have explicitly addressed a lack of time as a barrier to healthy eating [29]. Where home food preparation interventions have sought to address time scarcity, they have sometimes done so by providing quick recipe ideas on cards or websites, such as the online cooking and nutrition resource ‘No Money No Time’ [30].

While everyone has the same number of hours in a day, time ‘poverty’ or ‘scarcity’ refers to more demands being placed on those hours [20,31]. These demands can come in the form of paid employment, domestic tasks, or caring duties [20]. Indeed, individuals with high demands on their time, such as parents of young children who are employed outside the home, have been shown to prepare food at home less frequently [23,27]. However, Southerton and Tomlinson highlight that the experience of ‘harriedness’, endemic in contemporary life, may go beyond this requirement to spend time on necessary tasks and extends to other aspects of time, such as the weakening of socio-temporal structures, where increasingly unfixed schedules for things like work and meal times make it difficult to coordinate activities with families and households, and ‘temporal density’, involving multitasking and the erosion of boundaries between discrete tasks [32]. Nevertheless, the impact of demands on time in the form of paid or unpaid work to the experience of harriedness remains important. In an analysis of various measures of ‘time intensity’, including multitasking and task switching and their association with self-reported feelings of being rushed, it was found that the strongest predictor of feeling rushed was time spent on work [33].

Understanding what a lack of time means, practically, may be helpful in understanding whether different home food preparation interventions might be expected to work. It is also worth exploring whether making the desired change to more time spent on foodwork might be expected to lead to unintended consequences, depending on how individuals accommodate this new demand on their time and where they draw time from. To explore this, foodwork must be examined in conjunction with other daily activities.

Compositional data analysis is a technique that has recently been applied to the study of health behaviours, such as physical activity [34,35]. This approach construes a 24-h time budget as a composition made up of different activities, or parts, and takes into account some key properties of time: that time is bounded, and that budgeting time involves trade-offs between different activities.

While compositional data analysis has been applied in the field of nutrition to explore the nutritional composition of diets [36], it has not yet been applied to time spent on food practices in the context of other daily activities. The aim of this study was to use time-use diaries to explore the cross-sectional relationship between the extent of engagement with foodwork and the structure of a 24-h time budget (i.e., how much time people spend on daily activities). In analysing this time budget, we examined some activities which are health-promoting, such as sleep and physical activity, and others which are necessary to social, personal, and economic wellbeing, such as work, socialising, and leisure.

We further identified differences in this relationship between population subgroups, looking at three dimensions which have been shown to impact both time use and foodwork: gender, economic activity, and the presence of children in the household [15,23,27,37,38].

## 2. Materials and Methods 

### 2.1. Data Source

This study presents a secondary analysis of the 2014–15 United Kingdom Time Use Survey (UKTUS) [39], a cross-sectional national survey of UK residents aged 8 years and over. Private addresses were randomly sampled from UK postcode sectors [40]. From a total sample of 11,860 addresses, 10,479 were eligible. Ineligible addresses included non-residential addresses, holiday homes, and vacant buildings. Within each eligible household, one individual was asked to complete a household demographic questionnaire. All individuals in included households were asked to complete an individual demographic questionnaire and two 24-h time-use diaries (one weekday and one weekend day). Of the 10,479 eligible households, 40.4% responded, meaning a household questionnaire was completed, along with an individual questionnaire and one or two diary days from at least one resident. The study was approved by the Research Ethics Committee of the Department of Sociology at the University of Oxford (2014_01_02_R1).

### 2.2. Time Use Diaries

Participants were asked to fill out a time-use diary for one weekday and one weekend day selected by the study team. Diaries started at 4 am and covered a full 24-h period. This period was divided into 10-min time intervals, and participants were asked to fill in a primary activity for each time interval. All responses were given in free text and coded by the study team using a priori activity codes [40].

### 2.3. Exclusion Criteria

As suggested by the UKTUS study team, diaries characterised by three ‘flags’ indicating poor quality were excluded. These flags were: having more than 90 min of missing time, reporting fewer than seven episodes of activity (i.e., seven changes between activity or location), and missing two or more of four basic activities (sleeping/resting, eating/drinking, personal care, and exercise/travel) [41]. We further excluded any diaries that did not report a full 24 h of eligible activity codes, or that reported zero minutes spent on sleep. Of the diary days that passed these quality checks, we randomly selected one day for each participant aged 16 years and over.

### 2.4. Definition of Exposure (Foodwork)

We summed daily time spent on foodwork (time spent on shopping for food, food preparation and management, or washing dishes) for each participant. We assigned participants to one of three foodwork categories based on the amount of foodwork they had reported: ‘no foodwork’ (no time spent); ‘some foodwork’ (below the median amount of time spent for those who engaged in foodwork); and ‘more foodwork’ (above the median amount of time spent for those who engaged in foodwork).

### 2.5. Definition of Outcome (Time-Use Composition)

Foley et al. provide an overview of the compositional data analysis paradigm in health research [34]. Briefly, compositional data are made up of mutually exclusive parts which sum to a whole, such as, in this case, 24 h [42]. Transforming time-use data into a composition requires classifying time spent into different categories, with each category representing a part of the composition. We partitioned each participant’s time-use diary into seven mutually exclusive activity sets (parts) based on the activity they had reported in each time interval.

Personal care (e.g., showering, grooming).Sleep (including time spent in bed sleeping or in bed while not doing another activity).Eating.Physical activity (including walking and active transport by foot or bicycle).Leisure screen time.Work (including paid work as well as unpaid domestic work, such as foodwork, housework, and care work).Socialising and hobbies not captured elsewhere.

The specific activities included in each part are described in Appendix A.

All participant time could be allocated to one of these parts. Time spent travelling was allocated to the activity it enabled, with the exception of active travel (by foot or bicycle), which was coded to physical activity. Our parts reflected an interest in activities that are important to physical health, such as sleep and physical activity, as well as activities that may be important for social, economic, or psychological wellbeing, such as work or socialising.

Compositional information is relative rather than absolute, with ratios between parts being the primary interest. Compositional data analysis has the advantage of taking into account the co-dependent nature of compositional data, such as minutes available in a day, but standard analysis techniques, such as regression, cannot be directly applied to compositional data [34]. In order to apply these techniques, a common approach is to transform and express compositions as log-ratio coordinates (generated, in this analysis, using an isometric log-ratio transformation [43]). Expressed in this form, compositions may be treated as either exposures or outcomes in statistical models. Coordinates may then be back-transformed into original units for interpretation.

Because log-ratio coordinates may not be applied to zero values, the presence of zero values in one or more parts of a composition prohibits the use of compositional data analysis. Zeros in compositional data may be theorised as either ‘rounded’, representing a small nonzero value that falls below some detection limit, or ‘essential’, meaning a true zero and representing the complete absence of that part in the composition. Rounded zeros have been dealt with by imputing small nonzero values to replace them, but essential zeros remain a core challenge for compositional data analysis [44].

For this analysis we treated zeros as rounded, replacing zeros with small values under 10 min by drawing time from other parts to create imputed compositions. To do so, we used the log-ratio data augmentation algorithm function included in the R package zCompositions, which is a Markov Chain Monte Carlo algorithm and allows for the estimation of values below the detection threshold, while maintaining the relative structure of the data [45].

Our parts were defined in such a way that it seemed likely that most participants would spend at least a small amount of time engaging in each of the groups of activities, meaning reported zeros represented true small numbers. For example, a participant who had recorded no time spent socialising may still have greeted family members or colleagues or conversed with a supermarket cashier.

### 2.6. Covariates

Covariates were self-reported age, gender, economic activity (as defined by the Office for National Statistics: economically active, i.e., in paid employment or actively seeking work, or economically inactive [46]), occupational class (based on current or most recent employment using the three-class version of National Statistics Socio-Economic Classification [47], or not applicable for those who had never been in paid employment), age at leaving full-time education, and presence of children under the age of 16 in the home, as well as diary day type (weekend day or weekday) for the selected diary.

### 2.7. Analysis

We described the socio-demographic characteristics and the median time spent on foodwork for the whole sample and in each foodwork category. We conducted chi-square tests or one-way ANOVAs to identify statistically significant differences in the socio-demographic characteristics of each foodwork category. We then described the pattern of zero values in the time-use composition. All subsequent analyses were performed on the imputed compositions.

Because compositional data are constrained to sum to a whole, the values such data can take are bounded. As a result, they operate in a subset of real sample space known as the simplex [42]. As noted above, many analysis techniques traditionally employed in health research (in particular, linear regression) cannot be directly applied to compositional data, because such techniques assume that data are operating in real sample space [34]. In order to apply such techniques to compositional data, the data are transformed so that they operate in real space for which several approaches have been developed. For this analysis, we applied an isometric log-ratio (ilr) transformation to the data [43,48]. This transformation uses orthonormal bases to produce a set of ilr coordinates numbering one fewer than the number of parts. Each coordinate takes the form of a ratio between one part and another part or the geometric mean of several parts, in this case, sleep: personal care; eating: geometric mean of sleep and personal care; physical activity: geometric mean of sleep, personal care and eating; and so on [49].

In order to test for differences between time-use compositions for participants reporting no foodwork, some foodwork, and more foodwork, we followed the procedure suggested by Martìn-Fernandez and colleagues to interpret differences between groups of compositional data [49].

First, we used a multivariate analysis of variance (MANOVA) applied to the ilr transformation of the composition to determine whether the three groups differed [49]. We checked the assumptions of the MANOVA as recommended, using a multivariate goodness of fit test (from the R package compositions [48] developed based on Aitchison’s recommendation [42]) to verify the normality of residuals and a visual inspection of a dendrogram to verify the homogeneity of variances and covariances [48,49]. These tests suggested the assumptions of the MANOVA were met.

Second, if the results of the MANOVA suggested rejecting the null hypothesis of equality of means between the three groups of compositions, we used a Hotelling’s *T*-squared test, the multivariate generalisation of a standard *t*-test, to determine which pair of groups—none and some, or some and more—were different [49]. We chose not to analyse the third potential pair, none and more, as being less conceptually meaningful than the other pairs and, therefore, yielding results that would be difficult to interpret.

Third, where differences between two groups were detected, it was necessary to determine which of the individual parts differed. We estimated adjusted compositional means (i.e., adjusted for all covariates: age, gender, economic activity, occupational class, age at leaving education, presence of children in the household, and diary day type) for each group [34]. To do so, we created linear regression models with the ilr coordinates as outcome variables and the categorical foodwork variable as the exposure, along with the other covariates. Using the R package lsmeans [50], we estimated the adjusted mean ilr coordinate value for each of the six ilr coordinates. We did this separately for each foodwork category (none, some, and more), generating a complete set of six ilr coordinates for each category. Finally, we back-transformed these ilr sets, using the same ilr partitioning system, to obtain the adjusted compositional means for each foodwork category (none, some, and more).

Finally, we calculated the log-ratio differences in adjusted compositional mean between both pairs of groups: none vs some and some vs more. Log-ratio differences are log-transformed ratios, where the numerator is the model-adjusted minutes per day spent on a given part in a given group of participants, and the denominator is the model-adjusted minutes per day spent on the same part in another group of participants. For example, this could be the model-adjusted time spent sleeping in participants who do some foodwork compared to the model-adjusted time spent sleeping in participants who do no foodwork. In order to determine whether the difference in time spent was significant at the critical level, we constructed confidence intervals for each part using a bootstrap technique [49]. Confidence intervals that crossed zero indicated that there was no between-group difference for this part.

We entered interaction terms into the Hoteling’s *T*-squared models to determine whether the relationship between foodwork and time-use composition differed by gender, employment status, or presence of children in the home. Where the interaction term was significant, we stratified the sample and performed the analysis again for each subgroup, creating estimates for, for example, men and women separately.

For this analysis we used the open source software R (Version 3.6.1, R Foundation for Statistical Computing, Vienna, Austria,) and a number of bespoke packages for the analysis of compositional data, including Hotelling, lsmeans, Compositions, zCompositions, and robCompositions. Throughout this analysis we adjusted the critical level (0.05) in proportion to the number of groups analysed using the Bonferroni correction in order to prevent the artificial increase of the Type I error rate, as suggested by Martìn-Fernandez and colleagues [49]. This resulted in a critical level of 0.017, which was applied throughout.

## 3. Results

### 3.1. Sample Characteristics

The full data set consisted of 16,533 time-use diaries from 8274 participants. Of these, 23 diaries failed general quality checks, and 5005 diaries failed checks specific to this analysis (4988 reporting less than 24 h and 17 reporting no sleep). Of these valid diaries, 1182 were filled out by those aged under 16 years. After applying these exclusion criteria, we randomly selected one diary day from each participant, creating an analytic sample of 6143 diaries from 6143 participants.

Table 1 describes the characteristics of the analytic sample by foodwork category. Among participants who reported doing foodwork, the median amount of time spent on foodwork was 70 min. Participants doing less than 70 min of foodwork per day were, therefore, assigned to the ‘some’ foodwork category, with participants doing 70 min or more assigned to the ‘more’ foodwork category.

Participants in the higher foodwork categories were significantly older and more likely to be women than participants in the lower foodwork categories. Economically inactive participants—a group dominated in this sample by retired individuals—were over-represented in the more foodwork category. Meanwhile, participants who were still in full-time education were over-represented in the no foodwork category. Weekdays were slightly over-represented in the some foodwork category, perhaps reflecting shorter but more regular foodwork on days when participants were at work or school, while more foodwork is slightly more common on weekend days.

### 3.2. Differences between Time-Use Compositions Across Foodwork Categories

All analyses were performed on the imputed compositions, where zero values were replaced with small nonzero values. Patterns of zeros in the time-use composition are reported in Appendix B. After adjusting for covariates, there was a statistically significant difference in time-use composition between those reporting no foodwork, some foodwork, and more foodwork. The Hotelling’s *T*-squared test further suggested there was a statistically significant difference in time-use composition between both pairs of groups: no foodwork and some foodwork, and some foodwork and more foodwork. 

The model-adjusted compositional means for each part, presented separately for those reporting no foodwork, some foodwork, and more foodwork are shown in Figure 1. Symbols indicate a statistically significant log-ratio difference between foodwork categories for each part (*p* < 0.017).

The numerical values underlying Figure 1, Figure 2 and Figure 3 are in Appendix C.

With higher amounts of foodwork, more time was spent on work (a part which includes foodwork but also all other forms of work, both paid and unpaid), with participants who did more foodwork spending 102 min more on work than those who did some foodwork, and 137 min more on work than those who did no foodwork. Meanwhile, less time was spent on sleep. 

Relative to participants who did some foodwork, participants who did no foodwork spent more time eating (20 min, see Appendix C) and less time on physical activity (3 min) and watching screens (20 min). Meanwhile, participants who did more foodwork spent less time on personal care (12 min) and socialising and hobbies (15 min) relative to participants who did some foodwork.

### 3.3. Effect Modification

A statistically significant interaction (*p* < 0.017) was found for gender and economic activity in the association between foodwork and time-use composition but not for the presence of children in the household. The results of stratified analyses are presented in Figure 2 and Figure 3.

Figure 2 shows that women who did more foodwork spent less time on personal care and socialising and hobbies, which was not the case for men. Further, while both men and women who did more foodwork spent more time on work overall, women in all foodwork categories spent more time on work. This difference was smaller, at 6 min, between men and women who did no foodwork but larger, at 61 min, between men and women who did more foodwork (see Appendix C).

Figure 3 shows that both economically active and inactive participants spent more time on work in the higher foodwork categories. Economically active participants spent more time on work overall, as expected. However, the difference between economically active and inactive participants narrowed with increasing time spent on foodwork: in the no foodwork category, economically active participants spent 207 min more on work than economically inactive participants, while in the more foodwork category, economically active participants spent only 18 min more.

## 4. Discussion

### 4.1. Main Findings

This study explored the cross-sectional relationship between time spent on foodwork and the structure of a daily time budget. More time spent on foodwork was associated with less time sleeping and more time working. The latter may be partly explained by the inclusion of foodwork in work. However, for participants who did some foodwork versus no foodwork, the between-group difference in work is substantially larger (82 min, see Appendix C) than the difference in the geometric mean of time spent on foodwork (28 min, see Table 1). This suggests that participants who did some foodwork also did more of other types of work. Between participants who did more versus some foodwork, between-group differences in time spent on work were still larger than in time spent on foodwork (102 vs. 87 min), but the difference was less substantial. Given the lack of adjustment for covariates and the absence of the closing process for time spent on foodwork, these two groups may spend similar amounts of time on other types of work.

We also identified differences in time spent on foodwork and the structure of a daily time budget between population subgroups. The more foodwork category was dominated by women, while the no foodwork category was dominated by men. Economically inactive participants were over-represented in the more foodwork category.

Women who did more foodwork spent less time on personal care and socialising and hobbies, which was not the case for men. Women in all foodwork categories spent more time on work than their male counterparts. As time allocated to foodwork increased, this difference also increased from 6 to 61 min (see Appendix C).

Both economically active and inactive participants spent more time on work in the higher foodwork categories. The difference between economically active and inactive participants narrowed with increasing time spent on foodwork: in the no foodwork category, economically active participants spent 207 min more on work than economically inactive participants, while in the more foodwork category, economically active participants spent only 18 min more on work (see Appendix C).

### 4.2. Limitations of the Study

Because of our wish to look at the 24-h time budget as a composition, it was impossible to draw out time spent on foodwork and examine the remaining time in isolation, meaning foodwork is included in both exposure and outcome. However, foodwork (median 70 min/day) made up a relatively small proportion of daily time, and differences in time use were seen across several activity sets.

Time-use diaries tell us about substantive uses of time, but a substantial amount of intellectual labour goes into food planning and management [24,26,51,52]. This may occur alongside other tasks or in a fragmented way, potentially making participants less likely to record it in a time-use diary, meaning that time spent on foodwork may be underestimated here. Further, while time spent on foodwork is associated with diet quality [19], other factors could moderate this relationship, such as ingredients, kitchen equipment, cuisine, or skill.

This analysis uses one 24-h time-use diary from each participant. This one-day window may be less representative of participants’ usual time use than a longer diary, particularly for activities which participants engage in only infrequently [44]. However, the activities of interest in this analysis are relatively routine, and the conclusions of the analysis rely on measures of central tendency in the sample as a whole, rather than characterization of individual participants’ practices.

In existing research, participants who are more likely to experience time scarcity, such as single or working parents report lower frequency of preparing meals at home [20,23,27]. In contrast, in this analysis adults who lived in households with children were not over-represented in the lower foodwork categories, nor was the presence of a child in the house a significant modifier of the association between foodwork and time use. This difference could be attributable to our relatively rough measure of household structure, with participants being characterized based on the presence of children in their households. This may obscure substantial variation in household structure and responsibility for caring for children in the home. For example, participants with children who are single parents may be expected to spend substantially more time doing caring and housework than participants in dual-parent households. Meanwhile, some participants aged 16 and over may live in households with children, but they may be the siblings of these children rather than their parents. While children of different ages have been shown to contribute to housework [53,54], they generally spend less time on it than adults. Further analysis of foodwork and household structure could use a more differentiated measure to accommodate this variation.

Another explanation may be that the presence of children in the household leads to a different type or rhythm of foodwork, perhaps with more time spent on preparing snacks for children, but lower frequency of preparing what participants see as home-prepared meals. This hypothesis could be further explored through foodwork episodes combined with a more differentiated measure of household structure.

### 4.3. Implications of the Findings

These findings do not suggest that doing more foodwork is associated with less time spent on any single activity. Instead, the structure of a 24-h time budget varied by foodwork group across several activities, and this association further varied by socio-demographic characteristics.

In this sample, individuals who did more foodwork spent less time sleeping. Given the use of the compositional mean and the inclusion of daytime naps and all time spent in bed in the measure of sleep used, it is difficult to compare sleep time across different foodwork categories to guidelines, with both low and high amounts of sleep being detrimental to health [55]. However, an analysis of sleep in this sample using more conventional statistical methods concluded that the (arithmetic) mean time spent sleeping was in the recommended range, suggesting an epidemic of oversleeping in this sample is unlikely [56]. Given this, these results may suggest a less health-promoting pattern of sleep is associated with increased foodwork. It is plausible that sleeps acts as a ‘time reservoir’ from which time can be drawn to accommodate other activities, as has been concluded in studies on time use and physical activity [34,57].

Our results are consistent with existing work, which suggests that women do more foodwork and housework [26,58,59], and that, while women are increasingly in paid employment, they continue to do more than their share of work in the home [38]. This is of interest to our analysis in considering how women structure their time differently in order to accommodate the work they do.

Our findings suggest that gender continues to play a significant role in how foodwork is allocated. Past research suggests that even in households where the idea of domestic work as ‘women’s work’ is not explicitly endorsed, household members present alternative narratives to rationalise a gendered division of labour [26]. One such narrative is centred around health and budgeting: women feel that if they left their (male) partners to prepare meals they would not consider nutrition or cost [26]. As household gatekeepers they therefore feel obliged to take on the task themselves. These differences in the substantive use of time may mask further inequality in the intellectual labour implicit in foodwork: Cairns and Johnston discuss how their female participants would sometimes ask their (male) partners to go to the supermarket but would often frame this task themselves, preparing a list, shortening the list to only the items urgently required, and providing extensive instructions on the exact type of product required [60].

Our findings further show that this unequal responsibility for foodwork extends beyond time spent on foodwork itself to other daily activities, with less time being allocated to personal care, hobbies and socialising by women who do more foodwork than by men who do more foodwork. Practitioners who advocate or intervene to increase home food preparation must be careful to critically engage with gendered ideas around foodwork and responsibility for household health and budgeting.

In stratifying by economic activity, we found that time spent on work increased more substantially among economically inactive participants who did more foodwork than among economically active participants. Economically active participants were also under-represented in the more foodwork group, and unsurprisingly spent more time on work overall than those who were economically inactive. This may suggest that there is a limit to how much time participants are willing or able to spend working, whether this work is paid or unpaid.

Previous scholarship has discussed the interaction between time and income, suggesting that these two resources must be allocated in complementary ways: individuals who are more ‘time-poor’ may buy their way out of certain kinds of unpaid labour, such as working parents who pay for childcare [31,61]. Existing studies suggest this is true of foodwork: increased workforce participation and labour market hours worked by household managers (often women) are associated with increased frequency of consumption of pre-prepared meals, as well as increased expenditure on out of home food, often driving up overall food expenditure [62,63,64]. While home food preparation is advocated as an inexpensive strategy for eating healthily [65], in many households time and income poverty coexist, meaning that increasing home food preparation may be difficult. Given the increased financial costs associated with eating a healthier diet [66], these households may struggle to access healthy foods.

### 4.4. Future Research

While this cross-sectional analysis explores how participants who do more foodwork allocate their time differently than those who do less, it is not clear that these patterns would be replicated in the case of an individual increasing time spent on foodwork as a result of an intervention. Further work is required to determine what the effects on time use of such an intervention might be, and whether there are unintended consequences, such as health detriments due to a loss of time spent sleeping, an uneven allocation of additional work between genders, or a reduced effect for some households due to time and income poverty.

## 5. Conclusions

We found that time use varied extensively between participants who did more versus less foodwork. Participants who did more foodwork spent less time on sleeping, eating, socialising, and hobbies, while spending more time on paid and unpaid work, particularly when comparing participants who did any foodwork compared to none. This may have repercussions for physical health and broader dimensions of wellbeing.

Gender emerged as an important structuring factor in foodwork and time use. Women were over-represented in the category of participants doing more foodwork. In contrast to men, women who did more foodwork spent less time on personal care and socialising and hobbies.

While further work examining how time use changes as a result of a home food preparation intervention is certainly important, those who seek to encourage more home food preparation should be aware of the associations between time spent on foodwork and time spent on other activities and ensure their interventions guard against potential unintended consequences. Where home food preparation increases as a result of, for example, cooking classes or meal kit provision, re-allocation of time from other activities could be examined to determine what activities may be relinquished or curtailed in this process, and to ensure that gender imbalances are not being exacerbated.

## Figures and Tables

**Figure 1 nutrients-12-02280-f001:**
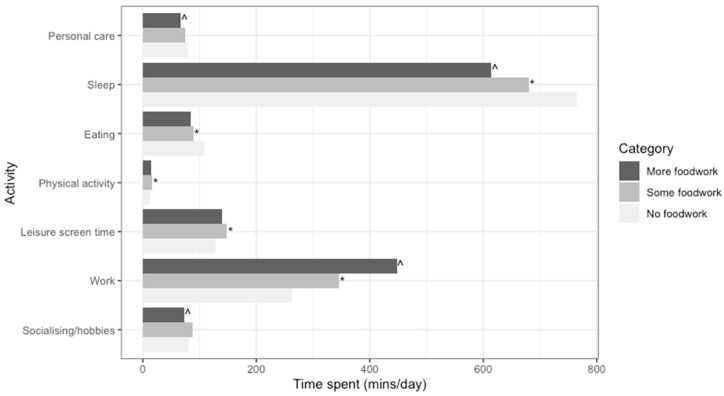
Model-adjusted ^a^ compositional means by foodwork category (*n* = 6143) ^a^ Adjusted for age, gender, employment status, education, occupation, presence of children, and diary day type. ^ Statistically significant log-ratio difference between more and some foodwork for this part. * Statistically significant log-ratio difference between some and no foodwork for this part.

**Figure 2 nutrients-12-02280-f002:**
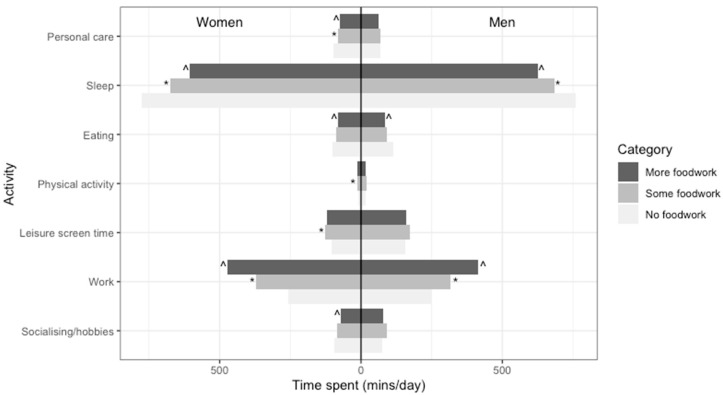
Model-adjusted ^a^ compositional means for men and women by foodwork category (*n* = 6143) ^a^ Adjusted for age, gender, employment status, education, occupation, presence of children, and diary day type. ^ Statistically significant log-ratio difference between more and some foodwork for this part. * Statistically significant log-ratio difference between some and no foodwork for this part.

**Figure 3 nutrients-12-02280-f003:**
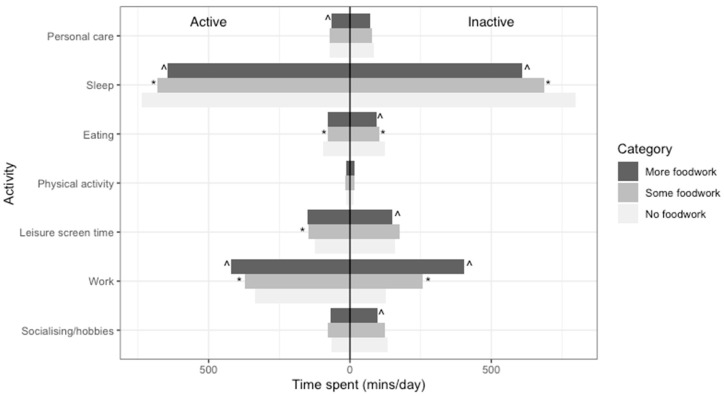
Model-adjusted ^a^ compositional means for economically active and inactive participants by foodwork category (*n* = 6143) ^a^ Adjusted for age, gender, employment status, education, occupation, presence of children, and diary day type. ^ Statistically significant log-ratio difference between more and some foodwork for this part. * Statistically significant log-ratio difference between some and no foodwork for this part.

**Table 1 nutrients-12-02280-t001:** Characteristics of analysis sample (n = 6143).

	No Foodwork	Some Foodwork	More Foodwork	Total		
Participants (n)	1455	2454	2234	6143		
Foodwork (mins/day):						
Median (IQR)	0 (0)	30 (20,50)	110 (90,150)	40 (10,90)		
Geometric mean	0	28.5	116.2	0	F	*p* value
Age (years, mean (SD))	41.8 (18.7)	46.8 (17.6)	53.2 (17.3)	47.9 (18.3)	191.4	<0.001
	n (%)				Pearson χ^2^	*p* value
Gender						
Men	958 (65.8)	1268 (51.7)	679 (30.4)	2905 (47.3)	475.65	<0.001
Women	497 (34.2)	1186 (48.3)	1555 (69.6)	3238 (52.7)		
Economic activity						
Economically active	982 (67.5)	1617 (66.2)	1103 (49.6)	3702 (60.5)	173.05	<0.001
Economically inactive	472 (32.5)	827 (33.8)	1120 (50.4)	2419 (39.5)		
Occupational grade						
Professional or managerial	444 (30.6)	920 (37.5)	736 (33.0)	2100 (34.2)	47.15	<0.001
Intermediate	381 (26.3)	695 (28.3)	629 (28.2)	1705 (27.8)		
Routine and semi-routine	407 (28.1)	611 (24.9)	616 (27.6)	1634 (26.6)		
Not applicable	217 (15.0)	226 (9.2)	251 (11.3)	694 (11.3)		
Children under 16 in household						
Yes	524 (36.0)	794 (32.4)	717 (32.1)	2035 (33.1)	7.21	0.027
No	931 (63.4)	1660 (67.6)	1517 (67.9)	4108 (66.9)		
Age at finishing full-time education						
Still in education	333 (22.9)	421 (17.2)	228 (10.2)	982 (16.0)	122.31	<0.001
16 or under	547 (37.6)	891 (36.3)	982 (44.0)	2420 (39.4)		
Over 16	575 (39.5)	1142 (46.5)	1024 (45.8)	2741 (44.6)		
Diary day						
Weekday	720 (49.5)	1304 (53.1)	1069 (47.9)	3093 (50.4)	13.64	0.001
Weekend	735 (50.5)	1150 (46.9)	1165 (52.2)	3050 (49.7)		

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
