# Peer review of "How Does Time Use Differ between Individuals Who Do More versus Less Foodwork? A Compositional Data Analysis of Time Use in the United Kingdom Time Use Survey 2014–2015"

_nutrients, 2020, doi:10.3390/nu12082280_

Round 1
Reviewer 1 Report
The paper presents a compositional approach to the analysis of data from a time use survey with a main focus on factors, which affect the time spent on activities related to eating.
First of all, I have to say that the contribution is well written and deserves publication. Even though the compositional methodology is nowadays not understood only as a way of dealing with closed data, but of any kind of data of a relative nature, the compositional approach is very appropriate here, since it allows for the study of the structure of the activities during a day. On the other hand, the methods are not clearly explained within the paper and that is the main issue I would like to ask the authors to fix.
My specific comments are as follows:
- in the whole paper, the references should be given before punctuation marks.
- p. 4 - "In order to apply these techniques, a common approach is to transform and express compositions as log-ratio coordinates." - Which coordinate system does the authors mean? How looks the transformation and how to transform the results back to the original sample space? At least a proper reference needs to be given.
- p. 4 - "For the analysis we treated zeros as rounded, replacing zeros with small values under 10...." - How was the zero imputation done?
- p. 4, ch. 2.7 Analysis - it is not clear, which steps were done on raw data and which on log-ratio coordinates. Moreover, the used coordinate system needs to be specified.
- p. 5 - What is the main principle of compositional MANOVA, how it differs from the classical one? Were the assumptions of the method inspected (see [46], p. 241-242)?
- p. 5 - It is not clear, what means "estimated adjusted compositional mean", please explain it better or provide a reference.
- p. 5 - "This transformation produces a set of ilr coordinates..." - the description of balances is not clear needs to be rewritten.
- p. 5 - "Linear regression models were created..." - the whole paragraph is very unclear. Was the response formed by all the ilr coordinates in one model? Why was the described procedure needed? Is it somehow related to the zero values imputation or computation of the adjusted compositional mean? At which point (for which values of regressors) was the mean computed?
Author Response
Dear reviewer,
We thank you for your comments and suggestions, which have been helpful in clarifying the study presented here. Please find a detailed response to these comments below. The manuscript has been revised with these comments in mind.
*Author responses in italics
in the whole paper, the references should be given before punctuation marks.
This has been addressed.
- 4 - "In order to apply these techniques, a common approach is to transform and express compositions as log-ratio coordinates." - Which coordinate system does the authors mean? How looks the transformation and how to transform the results back to the original sample space? At least a proper reference needs to be given.
Isometric log-ratio coordinates were used in this analysis. This has been specified, with reference to Egozcue et al. 2003 where the approach is described:
In order to apply these techniques, a common approach is to transform and express compositions as log-ratio coordinates (generated, in this analysis, using an isometric log-ratio transformation [43]).
This section represents an introduction to compositional data to help readers understand the compositional parts being introduced. Further detail on the transformation is given in a later section (2.7 Analysis), which we have also tried to clarify in line with reviewer comments.
- 4 - "For the analysis we treated zeros as rounded, replacing zeros with small values under 10...." - How was the zero imputation done?
A reference has been added to Palarea-Albaladejo and Martín-Fernández 2015 in Chemometrics and Intelligent Laboratory Systems, which describes the R package zCompositions, created to facilitate imputation in compositional data, which we used for imputation. We have also altered the text as follows, and hope it provides greater clarity:
For this analysis we treated zeros as rounded, replacing zeros with small values under 10 minutes by drawing time from other parts to create imputed compositions. To do so, we used the log-ratio data augmentation algorithm function included in the R package Zcompositions, which is a Markov Chain Monte Carlo algorithm and allows for the estimation of values below the detection threshold while maintaining the relative structure of the data [44].
- 4, ch. 2.7 Analysis - it is not clear, which steps were done on raw data and which on log-ratio coordinates. Moreover, the used coordinate system needs to be specified.
The MANOVA, Hotelling’s T-squared test and the calculation of adjusted compositional means and log-ratio differences were all performed on the transformed data. The description of the ilr transformation process has now been placed before these steps, and it has been specified that they used the transformed data, which we hope clarifies the issue.
- 5 - What is the main principle of compositional MANOVA, how it differs from the classical one? Were the assumptions of the method inspected (see [46], p. 241-242)?
The ‘compositional MANOVA’ refers to a MANOVA performed on an ilr transformation of a composition. Conformity to the assumptions were checked as recommended by Martin-Fernandez, and details of the checks have been added to the text. The paragraph has been re-written as follows:
First, we used a multivariate analysis of variance (MANOVA) applied to the ilr transformation of the composition to determine whether the three groups differed [49]. We checked the assumptions of the MANOVA as recommended, using a multivariate goodness of fit test (from the R package compositions [48], developed based on Aitchison’s recommendation [42]) to verify the normality of residuals, and a visual inspection of a dendrogram to verify the homogeneity of variances and covariances [48,49]. These tests suggested the assumptions of the MANOVA were met.
- 5 - It is not clear, what means "estimated adjusted compositional mean", please explain it better or provide a reference.
- 5 - "This transformation produces a set of ilr coordinates..." - the description of balances is not clear needs to be rewritten.
- 5 - "Linear regression models were created..." - the whole paragraph is very unclear. Was the response formed by all the ilr coordinates in one model? Why was the described procedure needed? Is it somehow related to the zero values imputation or computation of the adjusted compositional mean? At which point (for which values of regressors) was the mean computed?
Thank you for these comments. We have re-written these paragraphs in an attempt to respond to them:
Because compositional data are constrained to sum to a whole, the values such data can take are bounded. As a result, they operate in a subset of real sample space known as the simplex [42]. As noted above, many analysis techniques traditionally employed in health research (in particular, linear regression) cannot be directly applied to compositional data, because such techniques assume that data are operating in real sample space [34]. In order to apply such techniques to compositional data, the data are transformed so that they operate in real space, for which several approaches have been developed. For this analysis, we applied an isometric log-ratio (ilr) transformation to the data [43,49]. This transformation uses orthonormal bases to produce a set of ilr coordinates numbering one fewer than the number of parts, with each coordinate taking the form of a ratio between one part and another part or parts [48].
[Description of Martin-Fernandez method, MANOVA and Hotelling’s t-squared test as previously]
Third, where differences between two groups were detected, it was necessary to determine which of the individual parts differed. We estimated adjusted compositional means (i.e. adjusted for all covariates: age, gender, economic activity, occupational class, age at leaving education, presence of children in the household and diary day type) for each group [34]. To do so, we created linear regression models with the ilr coordinates as outcome variables and the categorical foodwork variable as the exposure, along with the other covariates. Using the R package lsmeans [50], we estimated the adjusted mean ilr coordinate value for each of the six ilr coordinates. We did this separately for each foodwork category (none, some and more), generating a complete set of six ilr coordinates for each category. Finally, we back-transformed these ilr sets, using the same ilr partitioning system, to obtain the adjusted compositional means for each foodwork category (none, some and more).
Reviewer 2 Report
The article presents a sound study of the interdependence of time devoted to home food preparation and the time budget. Data are carefully handled, control variables are wisely chosen and advanced statistical methods are used. References are up to date and cover the whole spectrum of disciplines related to the article topic. The topic has social relevance and a high contemporary interest.
I first have a few minor comments and then a major one which I feel would result in a radical improvement of the article.
MINOR COMMENTS:
Line 169. Please provide a reference supporting “For this analysis we treated zeros as rounded, replacing zeros with small values under 10 minutes by drawing time from other parts using a log-ratio data augmentation algorithm”.
Line 285. Caption to Fig. 2 is for some unknown reason shorter than those for Figs. 1 and 3.
Line 411. Could the authors provide a couple of interesting examples of intervention, which would be worth checking for success (or checking for failure)?
Table A3. The caption is in the middle of a two page sample.
Line 119 in the reference list, item #46. Provide all publication details.
Reference list as a whole: Check usage of capital letters.
MAJOR COMMENT.
I do not remember having ever seen a part of a composition acting as a covariate of the whole composition. So, all I’ll say is out of my intuition. The intuition of the authors does not seem to be very different from mine, since in lines 306-309 they state “More time spent on foodwork was associated with more time working and less time sleeping. While this may be partly explained by the inclusion of foodwork in work, between-group differences in work are larger (82 and 102 minutes, see Appendix C) than differences in median time spent on foodwork (30 and 80 minutes, see Table 1).” In my opinion this may be not not only a case of likely spurious correlation, but also a case of unnecessarily unclear interpretation. The fact that the authors warn readers in this respect is indeed valuable.
The authors later state that this limitation is unsurmountable (lines 324-326) “Because of our wish to look at the 24-hour time budget as a composition, it was impossible to draw out time spent on foodwork and examine the remaining time in isolation, meaning foodwork is included in both exposure and outcome.” I think it is not. At first glance I do not see why taking the subcomposition of all other activities except foodwork should be a problem. Aitchison himself suggested the use of subcompositions to compare groups defined by structural zeros in another component (in this case foodwork) which is not very different from what the authors aim to do. As for the reexpression of the results back into in minutes, the amount of foodwork time is known for each respondent, so that time in minutes could be recovered by closing each individual subcomposition to the total non-foodwork minutes, rather than to 24 times 60 minutes.
I provide a very simple example. 4 individuals and 2 components (work-other). 2 individuals spend 5 hours in food preparation, 2 individuals spend no time at all. The first table is the table as the authors use it: Foodwork time is added to work time. Higher work time seems to be associated to higher foodwork time while the opposite is actually the case. individuals i2 and i4 look compositionally identical.
Individual Foodwork Work Other Total_for_closure;
i1 5 15 9 24;
i2 0 10 14 24;
i3 0 13 12 24;
i4 5 10 14 24;
This is the table as I suggest it. Now individuals i2 and i4 nicely show a reallocation between work and foodwork, while individuals i1 and i2 are an example of reallocation between foodwork and other. Now individuals i1 and i3 look similar, in the way they distribute activities unrelated to foodwork, i.e. the subcomposition. The total_for_closure column should make it possible to reexpress the composition back into time units.
Individual Foodwork Work Other Total_for_closure;
i1 5 10 9 19;
i2 0 10 14 24;
i3 0 13 12 24;
i4 5 5 14 19;
I suggest the authors try this alternative approach and consider it, in case it enhances reliability and interpretability of their findings.
Author Response
Dear reviewer,
We thank you for your comments and suggestions, which have been helpful in clarifying the study presented here. Please find a detailed response to these comments below. The manuscript has been revised with these comments in mind.
*Author responses in italics
MINOR COMMENTS:
Line 169. Please provide a reference supporting “For this analysis we treated zeros as rounded, replacing zeros with small values under 10 minutes by drawing time from other parts using a log-ratio data augmentation algorithm”.
A reference has been added to Palarea-Albaladejo and Martín-Fernández 2015 in Chemometrics and Intelligent Laboratory Systems, which describes the R package zCompositions, created to facilitate imputation in compositional data.
Line 285. Caption to Fig. 2 is for some unknown reason shorter than those for Figs. 1 and 3.
Thank you, this has been addressed. The notes explaining the markings on the figures should be the same throughout.
Line 411. Could the authors provide a couple of interesting examples of intervention, which would be worth checking for success (or checking for failure)?
Thank you for this comment. We have added this sentence to the end of the conclusion:
Where home food preparation increases as a result of, for example, cooking classes or meal kit provision, re-allocation of time from other activities could be examined to determine what activities may be relinquished or curtailed in this process, and to ensure that gender imbalances are not being exacerbated.
Table A3. The caption is in the middle of a two page sample.
The second part of the table has now been captioned Table A3 (cont.)
Line 119 in the reference list, item #46. Provide all publication details.
Publication details have been added.
Reference list as a whole: Check usage of capital letters.
Capitalisation has been made more consistent in the reference list: only the first word of an article title is capitalised.
MAJOR COMMENT.
I do not remember having ever seen a part of a composition acting as a covariate of the whole composition. So, all I’ll say is out of my intuition. The intuition of the authors does not seem to be very different from mine, since in lines 306-309 they state “More time spent on foodwork was associated with more time working and less time sleeping. While this may be partly explained by the inclusion of foodwork in work, between-group differences in work are larger (82 and 102 minutes, see Appendix C) than differences in median time spent on foodwork (30 and 80 minutes, see Table 1).” In my opinion this may be not not only a case of likely spurious correlation, but also a case of unnecessarily unclear interpretation. The fact that the authors warn readers in this respect is indeed valuable.
The authors later state that this limitation is unsurmountable (lines 324-326) “Because of our wish to look at the 24-hour time budget as a composition, it was impossible to draw out time spent on foodwork and examine the remaining time in isolation, meaning foodwork is included in both exposure and outcome.” I think it is not. At first glance I do not see why taking the subcomposition of all other activities except foodwork should be a problem. Aitchison himself suggested the use of subcompositions to compare groups defined by structural zeros in another component (in this case foodwork) which is not very different from what the authors aim to do. As for the reexpression of the results back into in minutes, the amount of foodwork time is known for each respondent, so that time in minutes could be recovered by closing each individual subcomposition to the total non-foodwork minutes, rather than to 24 times 60 minutes.
I provide a very simple example. 4 individuals and 2 components (work-other). 2 individuals spend 5 hours in food preparation, 2 individuals spend no time at all. The first table is the table as the authors use it: Foodwork time is added to work time. Higher work time seems to be associated to higher foodwork time while the opposite is actually the case. individuals i2 and i4 look compositionally identical.
Individual Foodwork Work Other Total_for_closure;
i1 5 15 9 24;
i2 0 10 14 24;
i3 0 13 12 24;
i4 5 10 14 24;
This is the table as I suggest it. Now individuals i2 and i4 nicely show a reallocation between work and foodwork, while individuals i1 and i2 are an example of reallocation between foodwork and other. Now individuals i1 and i3 look similar, in the way they distribute activities unrelated to foodwork, i.e. the subcomposition. The total_for_closure column should make it possible to reexpress the composition back into time units.
Individual Foodwork Work Other Total_for_closure;
i1 5 10 9 19;
i2 0 10 14 24;
i3 0 13 12 24;
i4 5 5 14 19;
I suggest the authors try this alternative approach and consider it, in case it enhances reliability and interpretability of their findings.
Thank you for this carefully considered comment. Our analysis draws on the approach employed by Foley et al. 2018 (https://link.springer.com/article/10.1186/s12966-018-0662-8), where the association between active travel and overall time use was explored. Active travel was included in one of the compositional parts, physical activity, as foodwork is included as a form of ‘work’ in this analysis.
We were interested in the approach suggested by the reviewer and agree that it would be preferable if non-foodwork time . While it is feasible to obtain the compositional means and close to non-foodwork time as the reviewer suggests, we believe this approach causes problems in interpreting differences between groups. Where one compositional part makes up the same proportion of a different whole, the groups may appear not to differ significantly (i.e. their log-ratio differences may not be statistically significant), while in reality the number of minutes spent may be substantially different.
To illustrate:
Time spent in hours
Individual
i1
i2
Foodwork
0
4
Sleep
6
5
Other
18
15
Non-foodwork
24
20
Time spent as a percentage of non-foodwork time
Individual
i1
i2
Sleep
25
25
Other
75
75
Non-foodwork
100
100
Because the log-ratio analysis to determine whether groups of compositions are different is performed on the proportions rather than the numbers after closure, i1 and i2 would appear to spend a portion of time on ‘sleep’ and ‘other’ that was not significantly different. This masks a substantial difference of 1 and 3 hours, respectively. This difference would be reflected in the compositional means closed to non-foodwork time as the reviewer suggests, but not in the log-ratio differences. Indeed, when we re-analysed the data as suggested, a difference of 50 minutes in time spent sleeping between two groups registered as a non-significant log-ratio difference. Other smaller differences are more difficult to interpret.
However, we do note that, when excluding foodwork from the ‘work’ part, the difference in time spent on work between those who do no foodwork and some foodwork remains significant, while the difference between those who do some foodwork and more foodwork becomes non-significant. While, for the reasons described above, this is difficult to interpret, we have tried to word our conclusions more conservatively as follows:
This study explored the cross-sectional relationship between time spent on foodwork and the structure of a daily time budget. More time spent on foodwork was associated with less time sleeping and more time working. The latter may be partly explained by the inclusion of foodwork in work. However, for participants who did some foodwork versus no foodwork, the between-group difference in work is substantially larger (82 minutes, see Appendix C) than the difference in the geometric mean of time spent on foodwork (28 minutes, see Table 1). This suggests that participants who did some foodwork also did more of other types of work. Between participants who did more versus some foodwork, between-group differences in time spent on work were still larger than in time spent on foodwork (102 vs 87 minutes), but the difference was less substantial. Given the lack of adjustment for covariates and the absence of the closing process for time spent on foodwork, these two groups may spend similar amounts of time on other types of work.
We have also changed the wording in the conclusion to reflect this nuance:
We found that time use varied extensively between participants who did more versus less foodwork. Participants who did more foodwork spent less time on sleeping, eating, socialising and hobbies, while spending more time on paid and unpaid work, particularly when comparing participants who did any foodwork compared to none. This may have repercussions for physical health and broader dimensions of wellbeing.
Round 2
Reviewer 2 Report
First of all I want to thank the authors for seriously considering my comments. Although in some respects they did not follow the path I suggested, the response and changes are satisfactory. The conclusions, the method and the results now fit nicely together. As a whole the revision and response has produced an interesting debate.
Just one minor comment:
l. 202: please check sentence "with each coordinate taking the form of a ratio between one part and another part or parts".
I wish the authors all the best with this and future research.
Author Response
Dear reviewer,
We thank you for your careful consideration of our manuscript and helpful comments. We have edited the sentence you highlight as follows, and hope its meaning is now clearer:
Each coordinate takes the form of a ratio between one part and another part or the geometric mean of several parts, in this case, sleep: personal care; eating: geometric mean of sleep and personal care; physical activity: geometric mean of sleep, personal care and eating; and so on [49].